# Combining Mineral Amendments Improves Wheat Yield and Soil Properties in a Coastal Saline Area

**Jishi Zhang** [1],[†] ID **, Xilong Jiang** [1],[†]**, Qi Miao** [1]**, Botao Yu** [2]**, Liming Xu** [3] **and Zhenling Cui** [1],*

[1]   Center for Resources, Environment and Food Security, China Agricultural University, Beijing 100193, China; 15600660918@163.com (J.Z.); jiangxilong0513@126.com (X.J.); 18255025980@163.com (Q.M.)
[2]   Seed Multiplication Farm in Kenli County, Dongying 257500, China; kenliliangzhongchang@126.com
[3]   Agricultural High-Tech Industry Demonstration Area of the Yellow River Delta, Dongying 257300 China; ngqjfj@163.com
*   Correspondence: cuizl@cau.edu.cn; Tel.: +86-10-62733454; Fax: +86-10-32731016
†   These authors contributed equally to this work.

**Abstract:** Certain minerals possess structures that convey properties which improve soil quality; however, their application in coastal saline areas has been poorly studied. In this study, we explored the effects of combining mineral amendments on the improvement of wheat yield and soil properties in a two-year field experiment in mildly saline coastal soil areas of the Yellow River Delta, China. Five mineral materials were combined into the following four treatments: zeolite + rock phosphate (ZP), zeolite + silica calcium soil conditioner (ZC), vermiculite + rock phosphate (VP), and vermiculite + medical stone (VS). For all treatments, combined mineral amendments increased wheat yield compared to the control, with similar increases in yield following treatment with VP (45.7%), ZP (43.5%), and ZC (43.6%), and a significantly smaller increase following VS treatment (26.3%). These increases in grain yield were attributed to larger dry matter accumulation and higher grain numbers per ha. Compared to the control, ZP and ZC application substantially reduced soluble magnesium (Mg) and sodium (Na) contents, electrical conductivity (EC), and sodium adsorption ratio (SAR), and increased soil organic carbon (SOC) at a soil depth of 0–20 cm. VP application increased soil available phosphorus (P) by 34.7% and soluble potassium (K) by 69.3% at a soil depth of 0–20 cm. VS application slightly increased the SOC, total nitrogen (N), available P, and soluble K compared to the control. Overall, these results indicate that combining mineral amendments significantly increases wheat yield and improves soil properties in a saline area. Thus, we recommend the use of mineral amendments in saline coastal areas.

**Keywords:** coastal saline area; mineral materials; soil properties; wheat growth; yield

## 1. Introduction

The Yellow River Delta, the fastest growing river delta area, is a potential land resource in China, but food production in this area faces large challenges due to poor soil structure and salinity, which negatively impact soil quality, the absorption and transport of nutrient elements, and normal crop growth [1–5]. Irrigation is the main mode of salt leaching in coastal soil areas of the Yellow River Delta; however, poor soil properties, including low permeability and SOC, also limit the movement of salts to deep soil [1,6]. Thus, the application of amendments can improve soil properties and salt movement away from the plant root zone [7].

Traditional amendments such as gypsum [8], furfural residues [9], humic acid [7], and farmyard manure [10] improve soil physical and chemical properties, increasing crop growth in saline sodic soils [11]. For example, organic amendments can increase the flocculation of clay minerals and then

improve soil quality [12,13]. The addition of gypsum can improve the soil aggregate structure and soil permeability and reduce the $Na^+$ content due to $Ca^{2+}$ addition to the soil [14]. Some mineral amendment materials have special properties to improve soil conditions [15,16]. For example, zeolite, which is a microporous hydrated aluminosilicate mineral [17], has high cation exchange capacity (CEC) and a porous structure [16]. Zeolite can absorb many ions (e.g., Na), which remain loose and able to be exchanged by other ions [18]. Vermiculite, an aluminosilicate mineral similar to mica, has high CEC and specific absorption and exchange of cations [19]. Rock phosphate slowly releases phosphorus resources in low-phosphorus soils, promoting the absorption of phosphorus and increasing wheat yield [20]. Medical stone, an economical mineral material used in medical care and sewage disposal, has a porous structure and large surface area [15,21]. However, the applications of these minerals in coastal saline areas remain poorly studied.

Although the effects of a single mineral amendment strategy on soil properties have been studied [22], there are few reports on the effects of combining mineral amendments on soil properties and wheat growth in different growth stages. Thus, we explored the effects of combining mineral amendments on the wheat yield and soil physicochemical properties in a saline coastal area. We investigated wheat yield, yield components, accumulation of nutrient elements in wheat straw and grain following treatment with combined amendments, and the dynamics determining wheat stem number and dry matter at different growing stages in the saline coastal soil area. We also evaluated the impact of combining amendments on soil pH, organic C, total N, available P, soluble ions ($Ca^{2+}$, $K^+$, $Mg^{2+}$, and $Na^+$), electrical conductivity (EC), and sodium adsorption ratio (SAR).

## 2. Materials and Methods

### 2.1. Study Site

The field experiment was conducted in 2015–2016 and 2016–2017 in Kenli County (37°35′ N, 118°35′ E), Dongying, China. This region is characterized by a warm–temperate, monsoonal climate, with rainfall concentrated in summer. Daily average temperatures are typically near and below 0 °C between early December and late February in winter, and near 30 °C between June and August in summer. The annual precipitation is 550–600 mm, mainly in July and August. The major water resource for agricultural irrigation is the Yellow River. Figure 1 shows the precipitation and daily mean temperatures for the winter wheat growing seasons during 2015–2016 and 2016–2017; the total precipitation levels during the winter wheat growing seasons were 186.2 and 164.7 mm, respectively. Table 1 shows the physicochemical properties of the soil prior to sowing in 2015, in soil profiles at a depth of 0–40 cm. The field in which this study was conducted contains slightly saline soil, with an EC of approximately 1000 μS cm$^{-1}$ at a 1:5 ratio of soil to water.

**Table 1.** Chemical properties of the top 0–20 cm and 20–40 cm soil profile in the experiments and the five mineral amendment materials.

| Characteristics | Soil Depth (cm) | | Amendment Materials | | | | |
| --- | --- | --- | --- | --- | --- | --- | --- |
| | 0–20 | 20–40 | Zeolite | Vermiculite | Rock Phosphate | Medical Stone | Silica Calcium Soil Conditioner |
| pH | 7.68 | 7.67 | 9.64 | 9.04 | 9.13 | 8.34 | 9.57 |
| EC (μS cm$^{-1}$) | 917 | 692 | 146 | 531 | 3560 | 362 | 1991 |
| Organic carbon (g kg$^{-1}$) | 6.34 | 3.72 | - | - | - | - | - |
| Total N content (g kg$^{-1}$) | 0.81 | 0.58 | - | - | - | - | - |
| Available P (mg kg$^{-1}$) | 7.38 | 6.07 | 247 | 197 | 2169 | 279 | 354 |
| Soluble $Ca^{2+}$ (mg kg$^{-1}$) | 275 | 150 | 19.1 | 74.1 | 3931 | 191 | 3336 |
| Soluble $K^+$ (mg kg$^{-1}$) | 24.4 | 14.9 | 6.83 | 28.8 | 353 | 887 | 26.3 |
| Soluble $Mg^{2+}$ (mg kg$^{-1}$) | 96 | 47.2 | 51 | 14.8 | 353 | 66 | 175 |
| Soluble $Na^+$ (mg kg$^{-1}$) | 444 | 254 | 36.6 | 383 | 671 | 810 | 101 |

- not determined.

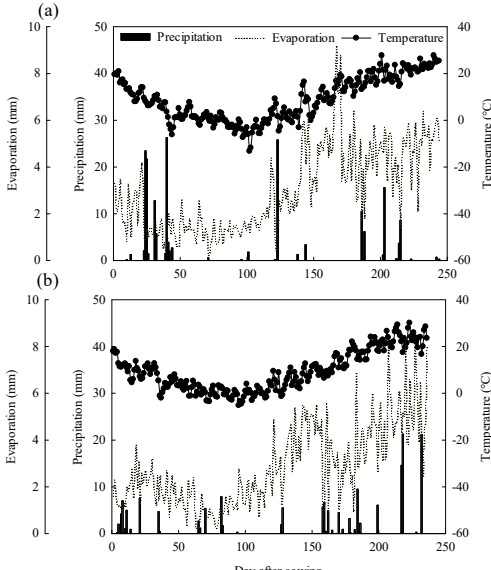

**Figure 1.** Precipitation and daily mean temperature during the winter wheat growing seasons (October–June) in 2015–2016 (**a**) and 2016–2017 (**b**) in Kenli County.

### 2.2. Mineral Amendment Materials

All amendments were microcrystalline (powder; <0.25 mm) and were provided by the Qing Da Powder Material Engineering Ltd. Co. (Zibo, Shandong, China). Zeolite is a natural aluminosilicate ore that includes a variety of nutrient elements [23]. Vermiculite is a natural, inorganic, nontoxic silicate mineral with good CEC and adsorption capacity [24]. The main component of rock phosphate powder is fluorapatite ($Ca_{10} (PO_4)_6 F_2$), which is slowly released from phosphate fertilizer [20]. Medical stone is another natural silicate mineral. Silica calcium soil conditioner is a product of calcium-rich amendment. All amendments were analyzed in the same way as soil. Table 1 shows the chemical properties of all amendments.

### 2.3. Experimental Design

The field experiment was arranged as a randomized block test design with six treatments and three replicates. The amendment treatments were the following: no amendment (control), zeolite + rock phosphate (ZP), vermiculite + rock phosphate (VP), vermiculite + medical stone (VS), and zeolite + silica calcium soil conditioner (ZC). All amendments were applied at a dosage of 1 Mg ha$^{-1}$. The wheat cultivar used in both seasons was "Jimai 22." The area of each plot was 300 m$^2$ (10 m × 30 m). The seeding rate was 600 seeds m$^{-2}$ at a row spacing of 16.7 cm, and the sowing dates were 14 October 2015 and 18 October 2016. The same amount of fertilizer was applied to each plot, and the total rates of N, $P_2O_5$, and $K_2O$ application were the same in both crop seasons: 180, 90, and 60 kg ha$^{-1}$, respectively. N fertilizer (urea, 46% N) was applied twice (60 kg ha$^{-1}$ before sowing, 120 kg ha$^{-1}$ at the stem elongation stage), and phosphate (monoammonium phosphate, 10% N and 50% $P_2O_5$) and K fertilizer (potassium sulfate) were applied as a basal fertilizer. Amendments and fertilizers applied before sowing were spread evenly on all plots and mixed into the top 20 cm of soil by deep plowing. Due to yearly water shortages in the Yellow River, the plots were irrigated only once (100 mm) at the stem elongation stage. Diseases and insect pests observed during the growing seasons of winter wheat were managed by applying the appropriate herbicides and insecticides.

### 2.4. Sampling and Laboratory Procedures

Meteorological data for the entire winter wheat growing seasons were obtained from the China Meteorological Data Network. Soil samples from a depth of 0–40 cm were collected after the wheat

harvest in 2016–2017 and before sowing in 2015–2016 and were tested for soil properties. Air-dried soil samples were sieved using a 2 mm sieve to test soil pH, EC, available P, soluble ions ($Ca^{2+}$, $K^+$, $Mg^{2+}$, and $Na^+$), and SAR, and using a 0.25 mm sieve to test SOC and total N. Soil pH (2.5:1 ratio of soluble $CaCl_2$ to soil, 0.01 mol $L^{-1}$) and EC (1:5, soil-to-water ratio) were tested using a pH meter and EC meter, respectively. Soil total N and SOC were tested using a vario MACRO cube elemental analyzer (Elementar, Langenselbold, Germany). Available P was measured using the ammonium molybdate–ascorbic acid method [25]. Soluble ions ($Ca^{2+}$, $K^+$, $Mg^{2+}$, and $Na^+$) were measured via extraction (5:1 water-to-soil ratio) and analyzed by titration ($Ca^{2+}$ and $Mg^{2+}$) and flame photometer ($K^+$ and $Na^+$) methods [8]. The SAR was calculated using Equation (1) [26], where $Na^+$, $Ca^{2+}$, and $Mg^{2+}$ are the concentrations of soluble ions in soil (mmol $L^{-1}$).

$$SAR = Na^+/[0.5(Ca^{2+} + Mg^{2+})]^{0.5} \tag{1}$$

Plant samples were collected at about 30 days of growth (seedling), and 150 days (reviving), 180 days (stem elongation), 200 days (flowering), and 240 days (maturity) after sowing each year to calculate the stem number and dry matter weight. Stem numbers were determined by counting the stems in a central row (1 m) within each plot. The dry matter weight was measured by reaping entire shoots (including straw and grain) within an area of 0.5 m in length and four rows in width in each plot and drying them to a constant weight at 60–65 °C in an oven. At maturity, the grain yield (water content: 14%) was calculated by harvesting plants in an area of 6 $m^2$ in each plot and threshing and drying the grain. The 1000 grain weight was calculated by weighing three groups of 500 grains each per plot. Grain number per spike was calculated by harvesting 30 spikes selected randomly per plot. To test the nutrient content, all plant samples (straw and grain) harvested at maturity were dried at 60–65 °C to a constant weight in an oven, digested with $HNO_3$–$H_2O_2$ in a microwave-accelerated reaction system (CEM Corporation, Matthews, NC, USA) [27], and determined by inductively coupled plasma–optical emission spectroscopy.

### 2.5. Statistical Analyses

The effects of amendment treatments, year, and their interactions on wheat yield, yield components, harvest index, and chemical composition of wheat straw and grain were determined by two-way analysis of variance (ANOVA). One-way ANOVA was used to analyze the effect of amendment treatments on the dynamics of dry matter accumulation, stem number per square, and soil properties (organic carbon, total N, available P, pH, EC, and soluble ions). Differences among treatments were detected using SAS software [28]. Multiple comparisons of average values ($p < 0.05$) were examined with Duncan's test at the 5% level.

## 3. Results

### 3.1. Wheat Grain Yield, Yield Components, and Dry Matter

Wheat grain yield was significantly affected by treatment, year, and their interaction (Table 2, $p < 0.001$). In both years, wheat grain yield in the ZP, VP, VS, and ZC treatments was significantly higher than in the control by 43.5%, 45.7%, 27.4%, and 43.6% ($p < 0.05$), respectively, due to increases in spike numbers (34.1–57.5%) and numbers of grains per spike (9.29–19.0%) compared to the control. Among all of the amendment treatments, grain yield was lowest in the VS treatment, with 6.61–11.3% lower grain weights than in other treatments. Compared to 2015–2016, the wheat grain yield was 15.6% lower in all treatments in 2016–2017, mainly due to reductions in spike number and grain weight by 12.4% and 11.0%, respectively (Tables 2 and 3).

In both years, the number of shoots was greater in all treatments than in the control throughout the growth season. Shoot numbers in the VS and ZC treatments were higher (averages: 555 and 549 $m^{-2}$, respectively) at the seeding stage, whereas the tiller capability (shoot number at stem elongation/shoot number at seeding) was greater in the ZP and VP treatments (averages: 3.65 and 3.99,

respectively). These results indicate a significant increase of 29.1–47.9% ($p < 0.05$) in shoot numbers under amendment treatments compared to the control at the stem elongation stage (Figure 2c,d). Lower spike numbers were observed in the ZC treatment at maturity than in all other amendment treatments, with a minimum percentage of stems and tillers of 39.0% (spike number/shoot number at stem elongation).

**Table 2.** Effects of the interaction between amendment treatments and year on wheat yield, yield components, and harvest index. ZP, zeolite + rock phosphate; VP, vermiculite + rock phosphate; VS, vermiculite + medical stone; ZC, zeolite + silica calcium soil conditioner.

| Year | Control | ZP | VP | VS | ZC |
|---|---|---|---|---|---|
| Yield (Mg ha$^{-1}$) | | | | | |
| 2015–2016 | 5.15c | 9.45a | 9.17a | 7.61b | 9.15a |
| 2016–2017 | 6.17c | 6.79b | 7.32a | 6.82b | 7.11ab |
| Spike number ($10^4$ ha$^{-1}$) | | | | | |
| 2015–2016 | 535c | 873ab | 979a | 839b | 769b |
| 2016–2017 | 539c | 818a | 711ab | 762ab | 671b |
| Grain number per spike | | | | | |
| 2015–2016 | 23.9d | 30.2b | 28.9c | 29.7bc | 34.1a |
| 2016–2017 | 29.8ab | 30.9ab | 31.9a | 29.1b | 30ab |
| Grain weight (g 1000$^{-1}$) | | | | | |
| 2015–2016 | 38.9b | 41.8a | 42.0a | 37.9b | 42.2a |
| 2016–2017 | 37.4ab | 36.5bc | 38.8a | 34.7c | 35.2c |
| Harvest index | | | | | |
| 2015–2016 | 0.39b | 0.45a | 0.43ab | 0.40b | 0.44ab |
| 2016–2017 | 0.43b | 0.44b | 0.47a | 0.48a | 0.44b |

Values represent the mean of three replicates. Different letters (a, b, and c) denote significant differences ($p < 0.05$) within a row as determined by Duncan's multiple comparison test.

**Table 3.** Wheat yield, yield components, and harvest index as affected by the treatments, year, and interaction between treatments and year. All data for the ANOVA come from Table 2.

| ANOVA | Sum of Squares | Degrees of Freedom | Mean of the Squares | F Value | *p* Value |
|---|---|---|---|---|---|
| Yield (Mg ha$^{-1}$) | | | | | |
| Year | 11.946 | 1.000 | 11.946 | 104.200 | <0.001 |
| Treatment | 28.785 | 4.000 | 7.196 | 62.773 | <0.001 |
| Year × Treatment | 12.544 | 4.000 | 3.136 | 27.355 | <0.001 |
| Spike number ($10^4$ ha$^{-1}$) | | | | | |
| Year | 11.946 | 1.000 | 11.946 | 104.200 | <0.001 |
| Treatment | 28.785 | 4.000 | 7.196 | 62.773 | <0.001 |
| Year × Treatment | 12.544 | 4.000 | 3.136 | 27.355 | <0.001 |
| Grain number per spike | | | | | |
| Year | 7.618 | 1.000 | 7.618 | 8.706 | 0.008 |
| Treatment | 87.307 | 4.000 | 21.827 | 24.946 | <0.001 |
| Year × Treatment | 85.401 | 4.000 | 21.350 | 24.401 | <0.001 |
| Grain weight (g 1000$^{-1}$) | | | | | |
| Year | 122.143 | 1.000 | 122.143 | 133.789 | <0.001 |
| Treatment | 53.742 | 4.000 | 13.435 | 14.717 | <0.001 |
| Year × Treatment | 27.753 | 4.000 | 6.938 | 7.600 | 0.001 |
| Harvest index | | | | | |
| Year | 0.007 | 1.000 | 0.007 | 13.485 | 0.002 |
| Treatment | 0.005 | 4.000 | 0.001 | 2.469 | 0.078 |
| Year × Treatment | 0.010 | 4.000 | 0.002 | 4.491 | 0.009 |

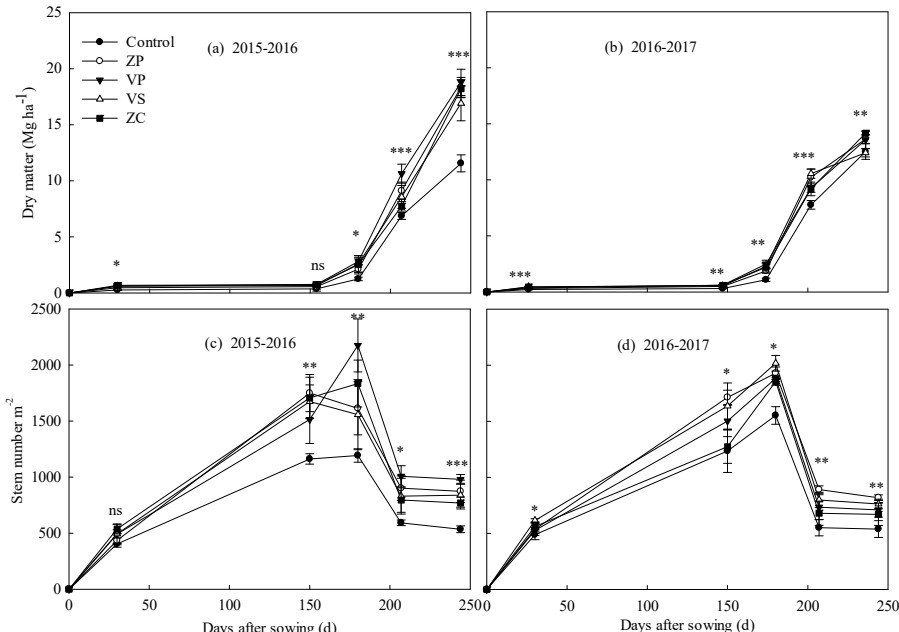

**Figure 2.** Dynamics of dry matter accumulation (**a**,**b**) and stem number per square (**c**,**d**) in response to different amendment treatments in 2015–2016 and 2016–2017. Vertical bars indicate ±SD of the mean. Different letters with each growth stage represent significant differences as determined by Duncan's test at the 5% level. ns, not significant; * significant at $p < 0.05$; ** significant at $p < 0.01$; *** significant at $p < 0.001$.

In both years, dry matter accumulation of the amendments was 15.6–133% higher compared to the control throughout the growth season (Figure 2a,b, $p < 0.05$). At maturity, the dry matter of the amendments was significantly higher than in the control, by 33.3%, 35.0%, 22.5%, and 35.0% in the ZP, VP, VS, and ZC treatments ($p < 0.05$), respectively. Increases in dry matter accumulation of 41.3%, 37.4%, 17.4%, and 72.9% were observed post-flowering. Across all treatments, the post-flowering dry matter was 48.9% and 29.2% of the total dry matter at maturity in 2015–2016 and 2016–2017, respectively.

### 3.2. Chemical Composition of Wheat Straw and Grain

In both years, the element concentrations in the grain and straw were significantly affected by the treatments ($p < 0.05$), with the exception of the Ca content in grain (Table 4). Compared with the control, all treatments significantly decreased the Na content by 25.5–67.8% ($p < 0.05$) in straw and 24.7–69.8% in grain ($p < 0.05$), and the Mg contents by 11.0–25.4% in straw ($p < 0.05$), except for VS in 2015–2016, and there was a better effect with ZP and ZC. Compared to the control, VP, VS, and ZC significantly reduced the Mg content of the grain by 5.00% ($p < 0.05$) in 2015–2016, but there was no significant difference under amendment treatments in terms of the Mg content of grain in 2016–2017, with the exception of ZP. The average Ca contents under ZP and ZC in wheat straw were significantly decreased by 18.3% and 15.0%, respectively, in 2015–2016 ($p < 0.05$) and by 16.4% and 21.7%, respectively, in 2016–2017 ($p < 0.05$) compared with the control. There was no significant difference among treatments and control for the Ca content of the grain. The P contents of straw under the ZP and VP treatments were significantly increased by 28.9% and 36.8% in 2016–2017 ($p < 0.05$), whereas no significant difference was observed in 2015–2016 across all treatments. The P content in grain was significantly increased by 5.86% ($p < 0.05$) with ZP in 2015–2016, and by 9.22% and 10.7% in 2016–2017 ($p < 0.05$) compared with the control, and the K content in straw was significantly increased by 12.4–32.0% ($p < 0.05$) across all amendment treatments except for VS in 2016–2017. VP and ZC significantly increased the K content of grain by 8.70% and 8.70%, respectively, in 2015–2016 ($p < 0.05$), and by 6.08% and 7.76%, respectively, in 2016–2017 ($p < 0.05$).

**Table 4.** Effects of amendment treatment and year on the chemical composition of wheat straw and grain.

| Cropping Years | Treatment | Wheat Straw | | | | | Wheat Grain | | | | |
|---|---|---|---|---|---|---|---|---|---|---|---|
| | | P | Ca | Mg | K | Na | P | Ca | Mg | K | Na |
| | | $g\ kg^{-1}$ | $g\ kg^{-1}$ | $g\ kg^{-1}$ | $g\ kg^{-1}$ | $mg\ kg^{-1}$ | $g\ kg^{-1}$ | $g\ kg^{-1}$ | $g\ kg^{-1}$ | $g\ kg^{-1}$ | $mg\ kg^{-1}$ |
| 2015–2016 | Control | 0.40a | 3.61ab | 1.72a | 20.0b | 1852a | 3.07b | 0.48a | 1.60a | 4.37b | 87.7a |
| | ZP | 0.45a | 2.95c | 1.34c | 24.3a | 623c | 3.25a | 0.48a | 1.55ab | 4.46b | 28.9b |
| | VP | 0.44a | 3.41b | 1.53b | 26.4a | 1010b | 3.10b | 0.48a | 1.52b | 4.75a | 36.6b |
| | VS | 0.36a | 3.81a | 1.57ab | 26.1a | 1024b | 2.73c | 0.48a | 1.52b | 4.53b | 32.0b |
| | ZC | 0.38a | 3.07c | 1.36c | 24.7a | 827bc | 3.09b | 0.46a | 1.52b | 4.75a | 26.5b |
| 2016–2017 | Control | 0.38b | 2.86a | 1.26a | 19.3b | 1721a | 3.47b | 0.41a | 1.65a | 4.77b | 77.2a |
| | ZP | 0.49a | 2.39cd | 1.01bc | 22.4a | 632d | 3.09c | 0.36a | 1.50b | 4.65b | 33.3c |
| | VP | 0.52a | 2.73ab | 1.06b | 23.2a | 1032c | 3.79a | 0.39a | 1.63a | 5.06a | 44.1c |
| | VS | 0.36bc | 2.53bc | 1.04b | 18.8b | 1283b | 3.84a | 0.40a | 1.68a | 5.00a | 58.1b |
| | ZC | 0.27c | 2.24d | 0.94c | 21.7a | 554d | 3.61b | 0.38a | 1.61a | 5.14a | 32.4c |
| Source of variation (*p* value) | | | | | | | | | | | |
| Year (Y) | | 0.8959 | <0.0001 | <0.0001 | <0.0001 | 0.6273 | <0.0001 | <0.0001 | 0.0014 | <0.0001 | 0.0561 |
| Treatment (T) | | 0.0003 | <0.0001 | <0.0001 | 0.0010 | <0.0001 | 0.0004 | 0.6074 | 0.0249 | <0.0001 | <0.0001 |
| Y × T | | 0.0818 | 0.0060 | 0.1577 | 0.0356 | 0.0230 | <0.0001 | 0.6495 | 0.0140 | 0.0785 | 0.0388 |

Values represent the mean of three replicates. Different letters (a, b, and c) denote significant difference ($p < 0.05$) within a column as determined by Duncan's multiple comparison test.

### 3.3. Soil Properties

Although the final soil pH (after wheat harvest in 2016–2017) was lower than the initial soil pH in all amendment treatments at a soil depth of 0–20 cm (7.68 before sowing in 2015–2016), there were no significant differences among treatments (Tables 1 and 5). Soluble ion contents were lower than the initial values at a 0–20 cm soil depth in all treatments, except for $K^+$ content. Compared to the control, the ZP and ZC treatments significantly reduced soluble $Mg^{2+}$ by 50.9% and 42.0% ($p < 0.05$), respectively, and reduced soluble $Na^+$ by 40.7% and 47.0% ($p < 0.05$), respectively. The soluble $Ca^{2+}$ content was significantly reduced by 31.8% in the ZP treatment ($p < 0.05$), and there were no significant differences among the other treatments. At a soil depth of 20–40 cm, soluble $Na^+$ ions were significantly reduced by 35.2–42.3% ($p < 0.05$) in all of the amendment treatments, with the exception of VS, and soluble $Mg^{2+}$ and $Ca^{2+}$ ion contents did not differ among treatments.

**Table 5.** Amendment treatment effect on soil organic carbon, total N, available P, pH, and soluble ions ($Ca^{2+}$, $K^+$, $Mg^{2+}$, $Na^+$) at soil depths of 0–20 and 20–40 cm after wheat harvest in 2016–2017.

| Soil Depth | Treatment | pH | Soil Organic Carbon (g kg$^{-1}$) | Total N (g kg$^{-1}$) | Available P (mg kg$^{-1}$) | Soluble Ions (mg kg$^{-1}$) | | | |
|---|---|---|---|---|---|---|---|---|---|
| | | | | | | Ca$^{2+}$ | K$^+$ | Mg$^{2+}$ | Na$^+$ |
| 0–20 cm | Control | 7.57a | 6.99b | 0.83c | 12.1b | 179a | 18.9c | 58.8a | 268a |
| | ZP | 7.58a | 8.70a | 0.99ab | 16.8a | 122b | 29.4ab | 28.9b | 159bc |
| | VP | 7.59a | 7.58b | 0.98ab | 16.3a | 188a | 32.0a | 51.5a | 216abc |
| | VS | 7.60a | 7.85ab | 0.87bc | 14.9a | 171a | 19.9c | 50.6a | 251ab |
| | ZC | 7.59a | 8.71a | 1.04a | 14.6ab | 142ab | 24.3bc | 34.1b | 142c |
| *p* value | | 0.7264 | 0.0173 | 0.0273 | 0.0223 | 0.0537 | 0.0084 | 0.0008 | 0.0569 |
| 20–40 cm | Control | 7.55c | 4.43b | 0.61b | 5.9c | 182a | 6.84b | 75.0a | 492a |
| | ZP | 7.58bc | 5.72a | 0.77a | 7.66bc | 208a | 7.74b | 53.7a | 312b |
| | VP | 7.54c | 5.45ab | 0.67b | 10.5a | 162a | 15.1a | 66.4a | 319b |
| | VS | 7.64b | 4.78ab | 0.64b | 8.79ab | 209a | 7.41b | 57.4a | 442a |
| | ZC | 7.72a | 5.59a | 0.69ab | 8.07abc | 164a | 7.89b | 60.5a | 284b |
| *p* value | | 0.0017 | 0.1060 | 0.0256 | 0.0246 | 0.5343 | 0.0017 | 0.3389 | 0.0133 |

Values represent the mean of three replicates. Different letters (a, b, and c) denote significant difference ($p < 0.05$) within a column as determined by Duncan's multiple comparison test.

The soil EC and SAR values in all amendment treatments were lower than in the control (Figure 3a,b). The EC values in the ZP (329 µS cm$^{-1}$) and ZC (319 µS cm$^{-1}$) treatments at a depth of 0–20 cm were significantly lower than in the control (580 µS cm$^{-1}$) ($p < 0.05$) and were lower than the initial EC value (917 µS cm$^{-1}$ before sowing in 2015); however, the SAR values only decreased by 37.3% in the ZC treatment. At a soil depth of 20–40 cm, the EC and SAR values were significantly lower in the ZP, VP, and ZC treatments than in the control by 22.6–33.6% and 30.6–35.2% ($p < 0.05$).

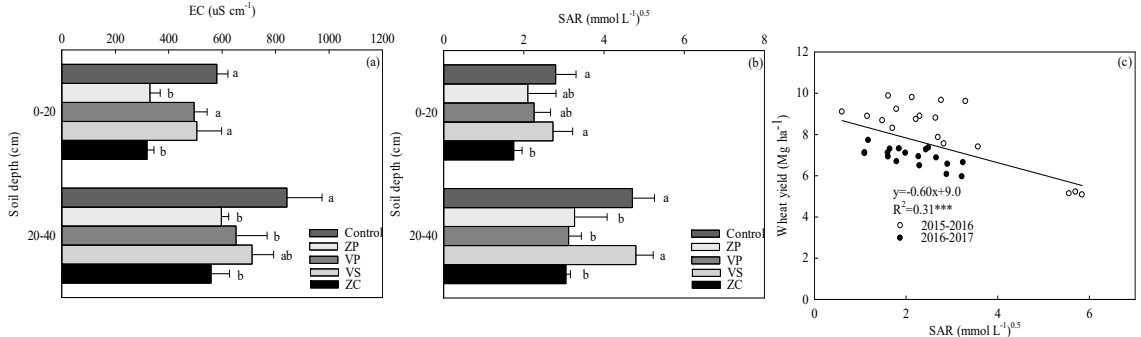

**Figure 3.** Treatment effect on soil electrical conductivity (EC) (**a**) and sodium adsorption ratio (SAR) (**b**) at soil depths of 0–20 and 20–40 cm after wheat harvest in 2016–2017, and the relationship (**c**) between wheat yield (Mg ha$^{-1}$) and soil SAR (0–20 cm soil layer). Values for the same year followed by the the same letter are not significantly different from each other ($p < 0.05$). *** Significant at $p < 0.001$.

Treatments significantly affected the SOC, total N, and available P (Table 5). The final SOC values were significantly higher than those of the control in only the ZP and ZC treatments, by 24.5% and 24.6% ($p < 0.05$) at a soil depth of 0–20 cm. The ZP, VP, and ZC treatments led to significantly higher total N than the control, by 19.3%, 18.1%, and 25.3% ($p < 0.05$), respectively, at a soil depth of 0–20 cm. The ZP and VP treatments led to significantly higher available P than in the control at the soil surface, by 38.8% and 34.7% ($p < 0.05$); the same trend was observed with soluble K, with increases of 55.6% and 69.3% ($p < 0.05$), respectively, compared to the control. At a soil depth of 20–40 cm, the VP treatment significantly increased available P and soluble K compared to the control, and available P was significantly higher following VS treatment, by 23.1% and 49.0% ($p < 0.05$) at soil depths of 0–20 and 20–40 cm, respectively.

## 4. Discussion

Our results showed that the application of a combination of mineral amendments significantly increased wheat yield by 27.4–45.8% and dry matter by 22.5–35.0%, which represents substantial gains compared to the results of Lu et al. [29] and Eroglu et al. [16]. There are many possible reasons for these results. First, poor soil structure, low SOC, and limited available nutrient content in saline coastal areas often leads to a reduction in nutrient absorption efficiency, limiting crop growth [10,30]. The properties of these mineral amendments (e.g., zeolite, vermiculite, and rock phosphate) improve soil structure and supply nutrients, then promote nutrient uptake and crop growth [12,16,20]. High biomass accumulation with amendment treatments increases the carbon return to the soil and improves soil fertility and nutrient supplies (e.g., P and K) in the soil [16,20,31,32]. For example, the porous structure of zeolite, which enhances soil water retention capacity, prevents nutrient loss and improves the growth and development of crops [22,32]; then, straw returns to the soil, increasing the SOC at the 0–20 cm soil depth, especially under the ZP and ZC treatments. Rock phosphate has a large Ca content, and zeolite and vermiculite have high CEC, so the ZP and VP treatments increase P and K contents by releasing P and allowing K to be replaced by Ca [17,24,31,33].

Second, combining mineral amendments reduced the soil EC and SAR at a depth of 0–20 cm and improved crop growth in this coastal area of the Yellow River Delta. The rock phosphate and silica calcium soil conditioner were rich in $Ca^{2+}$ (Table 1), which can replace $Na^+$ and $Mg^{2+}$ adsorbed onto soil colloids [34], so the ZP and ZC treatments significantly reduced the soil EC, soluble ions ($Na^+$ and $Mg^{2+}$), SAR, and Na and Mg contents in wheat straw and grain compared with those of the control (Table 5; Figure 3a,b). The decrease in Ca content in the surface soil was due to the replacement of Na ions adsorbed on soil colloids, resulting in low Ca content in wheat straw under the ZP and ZC treatments, which can be attributed to the high CEC of zeolite (Table 4) [16]. A previous study reported that zeolite has porous properties and confers a positive effect on CEC, which may promote ion substitution and reduce salt content [7,16,34,35]. The results of our study indicate that although vermiculite also has high CEC [24], the effects of the VP and VS treatments on soil salt reduction were poor in saline coastal soil, particularly with VS treatment. As the soil salt content decreased following the application of these combined mineral amendments, wheat development was promoted (Figures 2 and 3; Table 5).

Third, increased yield following the application of combined amendments contributed to higher numbers of grains per ha. This result is similar to that previously reported in the North China Plain, where an increase in wheat yield was mainly caused by an increase in spike numbers [36,37]. A previous study indicated that the early stage of wheat growth was more sensitive to salinity than the late stage of wheat growth [38]. In this study, our mineral amendments improved the growth and development of wheat during the vegetative growth stage and thus increased the numbers of grains per ha and dry matter accumulation at the pre-flowering stage (Figure 2a,b).

The increase in grain yield observed following treatment with combined mineral amendments was remarkable. We believe that this increase can be achieved in the vast majority of saline coastal soils if appropriate investments in this research field are made. For local farmers, net incomes following

the application of combination amendments must be considered along with the cost of amendment application. The costs of the ZP, VP, VS, and ZC treatments in this study were approximately 627 USD ha$^{-1}$ (cost per treatment) including material costs (537 USD ha$^{-1}$), traffic expenses (45 USD ha$^{-1}$), and labor services (45 USD ha$^{-1}$). Considering the potential increases in wheat yield and market price and the cost of improvement measures, the net incomes for the ZP, VP, and ZC treatments in this study potentially increased by about 239, 286, and 243 USD ha$^{-1}$, respectively, and that for the VS treatment decreased by 86 USD ha$^{-1}$. Although the application of these measures resulted in high yield and good income, most farmers do not actively adopt such technologies due to their operability, dosage, and labor costs. In the future, measures should be taken to optimize the relevant technologies and to encourage farmers to adopt them by means including technical adoption subsidies, forming a granular product to facilitate mechanization, providing adequate water resources through government support, and introducing other improvement measures (e.g., mulching methods) to achieve further yield increases. Meanwhile, amendment measures should be researched further to determine the duration of continuous application via weighing the effects of these measures on improving soil quality in saline coastal areas and the economic and environmental effects (e.g., potential heavy metal pollution).

## 5. Conclusions

In the coastal areas of the Yellow River Delta, poor soil quality and salinization restrains regional agricultural productivity. Our results demonstrated that the application of combined mineral amendments significantly increased wheat yield by 27.4–45.7% and dry matter by 22.5–35.0%. These gains were attributed partly to decreased soluble Na$^+$ content, EC, and SAR at the soil surface, and increased SOC and available P, thus increasing dry matter and wheat yield. In the future, decisions to apply these measures should consider their positive economic effects, improve mechanization, and reduce labor operations. These amendment measures are better than those typically used in wheat production due to their convenience of operation; grain production is anticipated to increase if these measures are further optimized through technology. We believe that the sustainable development of regional agriculture in saline coastal areas would be significantly improved through government support and entrepreneurial investment in these amendment measures and mechanized operation.

**Author Contributions:** Z.C. designed the research and supervised the project. J.Z., X.J. and Q.M. collected and analyzed the data. B.Y. and L.X. provided a platform and technical service for field experiments. J.Z. and X.J. wrote the manuscript.

**Funding:** The current work was funded the National Natural Science Foundation of China (U170621), Chinese National Basic Research Program (973, Program: 2015CB150400), Shandong and Israel International Cooperation in Scientific and Technological Cooperation Project: 2015SDY0105; Taishan Scholarship Project of Shandong Province (No. TS201712082); the Innovative Group Grant from NSFC: 31421092; Key Scientific and Technological Innovation Projects of Shandong Province (2017cxgx0311); and the National Key R D Program of China (Not 2017YFB0310801).

**Conflicts of Interest:** The authors declare no conflict of interest.

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
