# Peer review of "Combining Mineral Amendments Improves Wheat Yield and Soil Properties in a Coastal Saline Area"

_agronomy, doi:10.3390/agronomy9020048_

Round 1

Reviewer 1 Report

The authors have made a nive revision and addressed all my concerns.

I have only some very minor edits :

L41: flocculation "of" clay...

L43: "Ca2+ addition" instead of "add Ca2+"

L46-47: the sentence is unclear

L87: "were analyzed" instead of "are tested"

Table 3: in the p value column, please indicate <0.001 instead of 0.000

L180: delete "amendment"

L240: delete "and"

L244: what does "activate nutrient" mean?

L255-27: this sentence is unclear. What does "consumption of Ca ions" mean? What does "the reduction of Na ions" mean?

Author Response

L41: flocculation "of" clay...

We agreed with the comment and have replaced “flocculation clay minerals” with “flocculation of clay minerals”. (see line 41)

L43: "Ca2+ addition" instead of "add Ca2+"

Thank you for your comment. We agreed with the comment and replaced “add Ca2+” with “Ca2+ addition”. (see line 43)

L46-47: the sentence is unclear

We agreed with the comment and revised the sentence (see line 44-47). For example, zeolite, which is a microporous hydrated aluminosilicate mineral, has high cation exchange capacity (CEC) and porous structure. Zeolite can absorb many ions (e.g. Na), which remain loose and to be exchanged by other ions.

L87: "were analyzed" instead of "are tested"

Thank you for your comment. We replaced “are tested” with “were analyzed”. (see line 87)

Table 3: in the p value column, please indicate <0.001 instead of 0.000

Yes, you are right. We have revised the Table 3, and “0.000” with “<0.001”.

L180: delete "amendment"

We agreed with the comment and deleted “amendment”. (see line 180)

L240: delete "and"

We agreed with the comment and deleted “and”. (see line 239)

L244: what does "activate nutrient" mean?

Thank you for your comment. What we mean is that high biomass accumulation with amendment treatments increase carbon return to soil, and improve soil fertility and nutrient supplies (e.g., P and K) in the soil. And we have revised the description. (see line 243-244)

L255-27: this sentence is unclear. What does "consumption of Ca ions" mean? What does "the reduction of Na ions" mean?

Thank you for your comment and revised the sentence. The decrease in Ca content in surface soil was due to the replacement of Na ions adsorbed on soil colloids, thus resulting in low Ca content in wheat straw under ZP and ZC treatments, which can be attributed to the high CEC of zeolite (Table 4) .(see line 256-258)

Reviewer 2 Report

The corrections made according to the referee's comments are acceptable.

Author Response

Thank you very much for your approval of my revised manuscript.

This manuscript is a resubmission of an earlier submission. The following is a list of the peer review reports and author responses from that submission.

Round 1

Reviewer 1 Report

In this manuscript, the authors explored the influence of combining amendments on wheat yield and grain mineral concentrations, studying the dynamics of growth in the different phenological stages. They also evaluate the effects of combining amendments on physicochemical parameters of the soil, as pH, SAR, nutrient availability etc.

1) The objective of the research article is significant. Authors have judiciously gone through relevant literature before the planning of research, which is highly appreciable.

2) Methodologies adopted for investigations are standard and appropriate.

3) Studies also significantly reveal the potential of combining amendments on soil nutrient availability and the relationship  between wheat yield  and soil SAR. Therefore, authors are suggested to add few more measures (in materials and methods) besides to the control as chemical properties of the soil after addition of combining amendments.

I had  problems correctly reading table 3.

i) In Fig 1, the graphic bars of the precipitations, are not very evident and are confused with the graphic trace of evaporation.

ii) Table 3 should be reshaped, because it is unclear. Is the content of macronutrients (control and treated)  in grain and in straw the average over the two years? About the values of the years 2015-16 and 2016-17  what do they mean? are they control values?

The manuscript needs to be checked for punctuation marks/grammer that should either be added or  removed in case of full stops.

Author Response

Reviewer 1

1. In Fig 1, the graphic bars of the precipitations, are not very evident and are confused with the graphic trace of evaporation.

Thank you for your comment. We changed the formats of Fig 1.

2. Table 3 should be reshaped, because it is unclear. Is the content of macronutrients (control and treated) in grain and in straw the average over the two years? About the values of the years 2015-16 and 2016-17 what do they mean? are they control values?

We agreed with the comment and changed the format of Table 3. We showed the content of macronutrients in grain and in straw the average over the two years separately.

3. The manuscript needs to be checked for punctuation marks/grammer that should either be added or removed in case of full stops.

Thank you for your comment. We agreed with the comment and revised them in the new manuscript. We have made a revision in syntax and grammar by both native speakers of English. 

Reviewer 2 Report

The main problem with this manuscript is that authors mentioned that the soil is saline, but there is no sign of salinity. The EC of the soil is 0.917-0.6 dS/m  which is not consider as a saline soil. The authors should take out the concept of salinity from this manuscript. Instead, they should emphasise on the effect of amendments on improving the soil physical conditions or using the soil amendments as soil conditioner. 

The authors should use duncan test instead of LSD for comparision of the means. The authors should rewrite the introduction and discussion sections. They should show the significant of difference among means using alphabets. 

Author Response

Reviewer 2

1. The main problem with this manuscript is that authors mentioned that the soil is saline, but there is no sign of salinity. The EC of the soil is 0.917-0.6 dS/m which is not considered as a saline soil. The authors should take out the concept of salinity from this manuscript. Instead, they should emphasise on the effect of amendments on improving the soil physical conditions or using the soil amendments as soil conditioner. 

This is great point. We learned a lot. Current international classification of soil salinity was in terms of electric conductivity of saturated soil extract (ECe), and the ECe of slightly saline and moderately saline were 2-4 dS m-1 and 4-8 dS m-1 (Richards, 1954; Farifteh et al., 2008). However, we only tested the EC1:5, and the EC1:5 of the basics soil is 0.917-0.6 dS/m. In addition, many researchers have also done the relationship between ECe and EC1:5, and Lee et al. (2003) pointed out that ECe is approximately five times that of EC1:5. 

In the field experiment, the effect of salinity on crop growth was clearly observed in the whole growth season. Other research also reported the EC1:5 for saline soil in our study area. For example, Luo et al. (2016) considered the EC1:5 of the soil (1.00±0.01 dS m-1) as a saline coastal soil in the Yellow River Delta. 

Ultimately, in order to avoid confuse, we removed the saline coastal soil in the new version and change it to saline coastal area.

Luo, X. ; Liu, G. ; Xia, Y. ; Chen, L. ; Jiang, Z. ; Zheng, H.; Wang, Z. Y. Use of biochar-compost to improve properties and productivity of the degraded coastal soil in the yellow river delta, China. J. Soil Sediment. 2017, 17, 780-789.

Richards, L. A. Diagnosis and improvement of saline and alkali soils. USA Department of Agriculture handbook 60. USDA Government Printing Office, Washington, DC. 1954.

Farifteh, J.; Van der Meer, F.; Van der Meijde, M.; Atzberger, C. Spectral characteristics of salt-affected soils: A laboratory experiment. Geoderma. 2008, 145, 196-206.

Lee, S. H., Hong, B. D., An, Y., and Ro, H. M. Estimation of Conversion Factors for Electrical Conductivities Measure by Saturation-Paste and 1: 5 Water Extraction.  2003.

Lee, S. H.; Hong, B. D.; An, Y.; Ro, H. M. Estimation of conversion factors for electrical conductivities measured by saturation-paste and 1: 5 water extraction. Korean J. Soil Sci. Fertilizer. 2003, 36, 193-199.

2. The authors should use duncan test instead of LSD for comparision of the means. The authors should rewrite the introduction and discussion sections. They should show the significant of difference among means using alphabets.

Addressed. We have reanalyzed these data using the Duncan test and revised the introduction and discussion sections in the new manuscript.

Introduction:

We have revised the introduction and discussion. In the introduction, it is pointed out that poor soil quality and salinity together affect crop growth in the region, and we also supplemented the properties of mineral amendments, and so on. These mineral amendments have special structure properties, including high cation exchange capacity, porosity, rich in nutrients. They are mostly used in industry, as adsorbents, water retaining material, and little information was reported in agriculture, especially in saline soil area. 

In the discussion, we re-analyze the reasons, and also believe that the results of soil improvement should be primary. These mineral amendments improve soil structure, promote crop production, activate soil available nutrients, improve crop growth, and then form a virtuous cycle on the effect of soil quality and crop production.

Reviewer 3 Report

The paper by Zhang et al aims at investigating how the use of mineral amendments might improve wheat yield in saline soils. For this purpose, the authors carried out an in situ experiment and, after applying various amendments to the soil, they analyzed soil and plant properties over the next two years. The results are of novelty since they derive from an in situ experiment, which contrasts to traditional lab or greenhouse trials. The authors have also included a short economic analysis which is of interest. I have however some major concerns which should be resolved .

1.      The English level does not meet standards for publication as there are many syntax or grammar mistakes, especially in the discussion section. I therefore strongly suggest that a native speaker or a professional linguist proofreads the paper before the next submission.

2.      The introduction is confused. The authors state that some mineral amendments may accelerate salt movements to deep soil (L38-39). However, the amendments which are then presented and tested only aim at improving soil properties, without any impact on salt movement. It appears thus that these amendments are not selected combat to salinization but rather to improve soil fertility.

3.      The selection of the amendments is unclear. The authors give some characteristics (L40-47) which are somewhat disconnected with the salinization issue. The authors should better highlight why these amendments are believed to be efficient to improve crop yield in saline soils. Are there some other studies on this topic? How is the present study of novelty?

4.      L67-68: please, give the methodology used to determine soil and amendments properties.

5.      Statistical analyses are unclear. For Anova2, the authors should at least present Sum Sq, Df, F and P values for years, treatment and their interaction. In most of the results section, the authors used the “significant” term without showing the p value. This must be corrected before the next submission.

6.      The discussion is very confused. In my opinion, the authors should rewrite it because many explanation are rather strange (see specific comments below for examples) or too speculative.

7.      L273-286: It is unclear whether the amendments should be re-applied over time and if so, how often?

Specific comments

L75: powder: which size?

L83: “sowing date” instead of “sowing rate”?

L243-244: it is very strange that zeolite has a so high content of organic carbon. It would be useful -that the authors explain why.

L248: unclear. Why the improved P content in wheat should be due to increased wheat growth?

L252: The decreasing of what?

L258: nutrition???

L262: what is a “normal” soil?

Table 1: “organic carbon” instead of “soil organic carbon” as you also present the OC content of amendments.

Figure 2. Which statistical test? What is compared?

Author Response

Reviewer 3

1. The English level does not meet standards for publication as there are many syntax or grammar mistakes, especially in the discussion section. I therefore strongly suggest that a native speaker or a professional linguist proofreads the paper before the next submission.

Thank you for your comment. We have made a revision in syntax and grammar by at least two professional editors, both native speakers of English. For a certificate, please see: http://www.textcheck.com/certificate/uRuYxU.

2. The introduction is confused. The authors state that some mineral amendments may accelerate salt movements to deep soil (L38-39). However, the amendments which are then presented and tested only aim at improving soil properties, without any impact on salt movement. It appears thus that these amendments are not selected combat to salinization but rather to improve soil fertility.

Yes, you are right. Thank you for your comment. Although these mineral amendments don’t accelerate salt leaching, they do have special structure properties to improve soil fertility, absorb and exchange cations, and may be effective in improving saline soil. Therefore, we assumed that these mineral amendments could improve saline coastal soils, and our experimental results also confirmed our assumption. So we have made a revision in the introduction in the new manuscript.

3. The selection of the amendments is unclear. The authors give some characteristics (L40-47) which are somewhat disconnected with the salinization issue. The authors should better highlight why these amendments are believed to be efficient to improve crop yield in saline soils. Are there some other studies on this topic? How is the present study of novelty?

Thank you for your comment. Poor soil quality and salinity limited crop growth in saline coastal areas. The properties of these mineral amendments improve soil structure, as well as supply nutrients, and then promote nutrient uptake and crop growth, increase crop yield, increase carbon return to soil, improve soil fertility. For example, zeolite could improve soil quality in saline soil because its porous structure and high CEC can accommodate massive cation (e.g., Na+) so that these ions remain loose and be exchanged by other ions.

4.  L67-68: please, give the methodology used to determine soil and amendments properties.

Thank you for your comment. The methodology used to determine soil and amendments properties was same in sampling and laboratory procedures section of the new manuscript, and we have supplemented the description.(see line 89-90)

5.  Statistical analyses are unclear. For Anova2, the authors should at least present Sum Sq, Df, F and P values for years, treatment and their interaction. In most of the results section, the authors used the “significant” term without showing the p value. This must be corrected before the next submission.

We agree with the comment. We supplemented this ANOVA for all data of Table 2 (see Table 3). And we have added the p value in the results section.

6.  The discussion is very confused. In my opinion, the authors should rewrite it because many explanation are rather strange (see specific comments below for examples) or too speculative.

Thank you for your comment. We re-analyze the reasons, and also believe that the results of soil improvement should be primary. These mineral amendments improve soil structure, promote crop production, activate soil available nutrients, improve crop growth, and then form a virtuous cycle on the effect of soil quality and crop production. So we deleted some unclear or confusing descriptions, such as the effect of rock phosphorus supplying P and mineral amendments supplementing carbon sources.

7.  L273-286: It is unclear whether the amendments should be re-applied over time and if so, how often?

Thank you for your comment. Yes, you are right. However, it is not yet possible to determine the time for re-applied these measures, so we should further research the effects of these measures on improving soil quality in saline coastal soil area, economic and environmental benefits (e.g. potential heavy metal pollution) by continuous application.

Specific comments

L75: powder: which size?

Thank you for your comment. The size of all amendments was less than 0.25mm, which was added in this new version.

L83: “sowing date” instead of “sowing rate”?

Addressed. Yes, you are right. We replaced the “sowing rate” to be “sowing date

L243-244: it is very strange that zeolite has a so high content of organic carbon. It would be useful -that the authors explain why.

Addressed. This is a measurement error. The organic carbon content with all mineral materials is relatively low, and application of mineral amendments is not enough to cause changes in soil organic carbon. Therefore, we did not emphasize the effect of mineral materials on soil organic carbon, but increased the effect of growth of crops and the straw returned to the soil in the new manuscript. This point was also elaborated in the discussion.

L248: unclear. Why the improved P content in wheat should be due to increased wheat growth?

Thank you for your comment. The P content in saline coastal areas is low (Table 1), and P fertilizer applied to soil is easily immobilized. But some mineral amendments could promote P absorption in wheat. However, we also realize that this is not the most important reason for the increase in grain yield. We speculated that the reason might be that mineral amendments applied to soil improved soil structure, increased crop growth, and then promoted crop P uptake. 

L252: The decreasing of what?

Thank you for your comment. The decreasing with Ca content in wheat straw and soil surface with ZP and ZC treatments may be due to excessive consumption of calcium ions while reducing sodium ions, which was attributable to cation exchange capacity of zeolite (Table 4)

L258: nutrition???

Addressed. Yes, you are right. We replaced the “rich in calcium and phosphorus nutrition” to be “rich in calcium and phosphorus” (see line 271).

L262: what is a “normal” soil?

Addressed. Yes, you are right. We have realized that the word is inaccurate and have revised it.

Table 1: “organic carbon” instead of “soil organic carbon” as you also present the OC content of amendments.

Addressed. Yes, you are right. We replaced the “soil organic carbon” to be “organic carbon(see Table 1).

Fig 2. Which statistical test? What is compared?

Thank you for your comment. We used one-way ANOVA within five treatments in each growth stage, and different letters with each growth stage represent significant difference by Duncan test at the 5% level. Ns, not significant; *significant at p < 0.05; **significant at p < 0.01; ***significant at p < 0.001.
